# A serum microRNA classifier for the diagnosis of sarcomas of various histological subtypes

Naofumi Asano[1,2], Juntaro Matsuzaki [3], Makiko Ichikawa[4], Junpei Kawauchi[4], Satoko Takizawa[4], Yoshiaki Aoki[5], Hiromi Sakamoto[6], Akihiko Yoshida[7], Eisuke Kobayashi[8], Yoshikazu Tanzawa[8], Robert Nakayama[2], Hideo Morioka[2], Morio Matsumoto[2], Masaya Nakamura[2], Tadashi Kondo[1], Ken Kato [9], Naoto Tsuchiya[10], Akira Kawai[8] & Takahiro Ochiya[3,11]

Due to their rarity and diversity, sarcomas are difficult to diagnose. Consequently, there is an urgent demand for a novel diagnostic test for these cancers. In this study, we investigated serum miRNA profiles from 1002 patients with bone and soft tissue tumors representing more than 43 histological subtypes, including sarcomas, intermediate tumors, and benign tumors, to determine whether serum miRNA profiles could be used to specifically detect sarcomas. Circulating serum miRNA profiles in sarcoma patients were clearly distinct from those in patients with other types of tumors. Using the serum levels of seven miRNAs, we developed a molecular detector, Index VI, that could distinguish sarcoma patients from benign and healthy controls with remarkably high sensitivity (90%) and specificity (95%), regardless of histological subtype. Index VI provides an approach to the early and precise detection of sarcomas, potentially leading to curative treatment and longer survival.

[1] Division of Rare Cancer Research, National Cancer Center Research Institute, 5-1-1 Tsukiji, Chuo-ku, Tokyo 104-0045, Japan. [2] Department of Orthopaedic Surgery, Keio University School of Medicine, 35 Shinanomachi, Shinjuku-ku, Tokyo 160-8582, Japan. [3] Division of Molecular and Cellular Medicine, National Cancer Center Research Institute, 5-1-1 Tsukiji, Chuo-ku, Tokyo 104-0045, Japan. [4] Toray Industries, 6-10-1 Tebiro, Kamakura City, Kanagawa 248-0036, Japan. [5] Dynacom Co., Ltd., WBG E25 2-6-1 Nakase, Mihama-ku, Chiba 261-7125, Japan. [6] Division of Biobank and Tissue Resources, National Cancer Center Research Institute, 5-1-1 Tsukiji, Chuo-ku, Tokyo 104-0045, Japan. [7] Department of Pathology and Clinical Laboratory, National Cancer Center Hospital, 5-1-1 Tsukiji, Chuo-ku, Tokyo 104-0045, Japan. [8] Department of Musculoskeletal Oncology, National Cancer Center Hospital, 5-1-1 Tsukiji, Chuo-ku, Tokyo 104-0045, Japan. [9] Department of Gastrointestinal Medical Oncology, National Cancer Center Hospital, 5-1-1 Tsukiji, Chuo-ku, Tokyo 104-0045, Japan. [10] Laboratory of Molecular Carcinogenesis, National Cancer Center Research Institute, 5-1-1 Tsukiji, Chuo-ku, Tokyo 104-0045, Japan. [11] Department of Molecular and Cellular Medicine, Tokyo Medical University, 6-7-1, Nishishinjuku, Shinjuku-ku, Tokyo 160-0023, Japan. These authors contributed equally: Naofumi Asano, Juntaro Matsuzaki. Correspondence and requests for materials should be addressed to T.O. (email: tochiya@ncc.go.jp)

Bone and soft tissue tumors are rare diseases that arise from mesenchymal tissues and exhibit wide variation in histologic type and malignant grade. They are classified into three major categories: benign, intermediate, and malignant[1]. Benign tumors do not spread beyond the original site (i.e., they do not metastasize) and are not life-threatening. Intermediate tumors are defined as locally aggressive but nonmetastasizing or rarely (<5% of cases) metastasizing, and most cases require surgical resection for local control. On the other hand, the malignant tumors of bone and soft tissue known as sarcomas constitute an extremely rare heterogeneous group of cancers that account for 1% of adult and 15% of pediatric neoplasms; these cancers are highly metastatic and often life-threatening[1–3].

Despite recent advances in precision medicine and improvements in patient prognosis for several types of cancer, few advances have been made in the treatment of sarcomas. The development of molecular targeted therapy for sarcomas is hindered by the fact that these tumors are classified into more than 50 histological subtypes, and the number of cases included in studies has been limited[1,2]. The current 5-year survival rate of patients with sarcoma is approximately 50–60%, depending on histological grade and tumor depth, size, and stage[1,3–5]. Due to the low degree of public and professional awareness of these malignancies, bone and soft tissue sarcomas are often diagnosed at late stages[6,7]. Moreover, due to their rarity and diversity, and despite their classification into the aforementioned malignancy categories, the diagnosis of sarcomas remains difficult[8]. Hence, to improve the prognosis of patients with sarcoma, early detection is essential.

Currently, histopathological examination by specialists is the only method available for the accurate diagnosis of these tumors, and no biomarkers have been identified to date. To achieve early and accurate diagnosis of sarcoma, it will be necessary to develop rapid, noninvasive, and straightforward diagnostic methods. The ideal biomarkers for this purpose should have the capacity to detect sarcomas regardless of their histological subtype. To this end, it is necessary to develop a biomarker identification strategy based on a novel principle.

MicroRNAs (miRNAs) are a class of small noncoding RNAs that act as endogenous regulators of gene expression. The tissue-specific expression pattern of miRNAs is important for the precise regulation of cell differentiation and tissue development, and alterations in these processes are involved in the pathogenesis of cancer[9,10]. miRNA expression signatures are tumor type-specific and enable the classification of tumors by origin[11,12]. Certain subsets of miRNAs are aggressively secreted from cancer cells into the extracellular space via multiple mechanisms, including microvesicle (MV)-mediated pathways[13,14]. In light of these biological features of miRNAs, the profile of extracellular miRNAs, also known as circulating miRNAs, is expected to be cancer type-specific and reflect tumor origin[15,16]. Accordingly, the use of circulating miRNA profiles warrants further investigation as a strategy for overcoming the challenges associated with diagnosis of sarcomas.

The use of serum miRNAs as markers for cancer diagnosis has recently been explored[16]; however, no consensus has emerged regarding which miRNAs are most suitable for clinical application, nor is there any standard method for collection of serum samples or preparation and detection of miRNAs. The use of circulating miRNAs in the clinic requires large comprehensive analyses of several cancer types, using standardized platforms for serum miRNA collection and detection. We recently launched a national project in Japan, entitled Development and Diagnostic Technology for Detection of miRNA in Body Fluids. This project entails the comprehensive characterization of serum miRNA profiles in more than 10,000 patients with 13 types of human cancer, including sarcomas, using the same platform and technology to evaluate each sample. The ultimate goal of this project is the development of diagnostic methods for use in clinical examinations that are capable of detecting and classifying patients according to cancer type by analyzing the circulating miRNAs in single blood samples.

Here, we describe our findings regarding the identification of promising biomarkers for the diagnosis of sarcoma. In particular, we developed Index VI, calculated from the levels of seven serum circulating miRNAs, which could discriminate among the various histological subtypes of sarcoma. Our findings provide a solution to the daunting challenges associated with the diagnosis of sarcoma.

## Results

**Assay design.** We obtained 1002 serum samples derived from patients with bone and soft tissue tumors. After excluding five patients with uterine sarcoma, 24 patients who had undergone treatment before serum collection, and 76 low-quality samples, we analyzed the miRNA profiles in material from 897 patients, including serum samples from patients with malignant bone and soft tissue tumors (sarcomas) ($n = 414$), intermediate tumors ($n = 144$), and benign or nontumors ($n = 339$).

Samples from patients with sarcomas and benign tumors were divided into four cohorts: discovery, training, and validation sets for microarray analysis; and training set 2 for quantitative RT-PCR (qRT-PCR) (Fig. 1). Serum samples from 150 and 125 healthy volunteers were included in the training and validation cohorts, respectively. Clinical features, including age, sex, site of primary tumor, and TNM stage, did not differ significantly between the sets ($P > 0.01$ after Bonferroni correction for multiple testing) (Table 1). The distribution of histological subtypes of bone and soft tissue tumors is shown in Supplementary Tables 1 and 2. Age and sex distributions of malignant, benign, and healthy participants were matched in the discovery set, but not in the training and validation sets (Supplementary Table 3). Therefore, age- and sex-adjusted analysis was conducted after the identification of a diagnostic index, as described below.

**Profiles of serum miRNAs.** A total of 351 miRNAs with signal value $>2^6$ in more than 50% of the analyses in the malignant or benign tumor group in the discovery set were defined as abundant serum miRNAs. Unsupervised cluster analysis identified differences in circulating miRNA profiles between primary sarcomas ($n = 311$) and benign tumors ($n = 327$) (Fig. 2a). This effect was confirmed by principal component analysis (PCA) (Fig. 2b). Serum miRNA profiles could not distinguish tumors according to origin, histological subtype of sarcoma, or lineage differentiation, as evidenced by unsupervised cluster analysis (Supplementary Figure 1A) and PCA (Supplementary Figure 1B–D). These results suggested that circulating miRNA profiles of bone and soft tissue sarcomas are similar, irrespective of the histological subtype. This result is reasonable given that sarcomas share a common mesenchymal origin, and that serum miRNA profiles reflect the origins of human cancers[11].

**Construction of diagnostic models using serum miRNA profiles.** A total of 83 miRNAs exhibited high discrimination performance between sarcomas and benign tumors, based on leave-one-out cross-validation in the discovery set (Supplementary Table 4). The serum levels of these miRNAs were validated using the training cohort. In a cluster analysis, 33 of these 83 miRNAs were included in a cluster whose expression levels increased step-by-step with disease severity (i.e., lowest in healthy patients, moderate in benign tumors, and high in malignant tumors). On

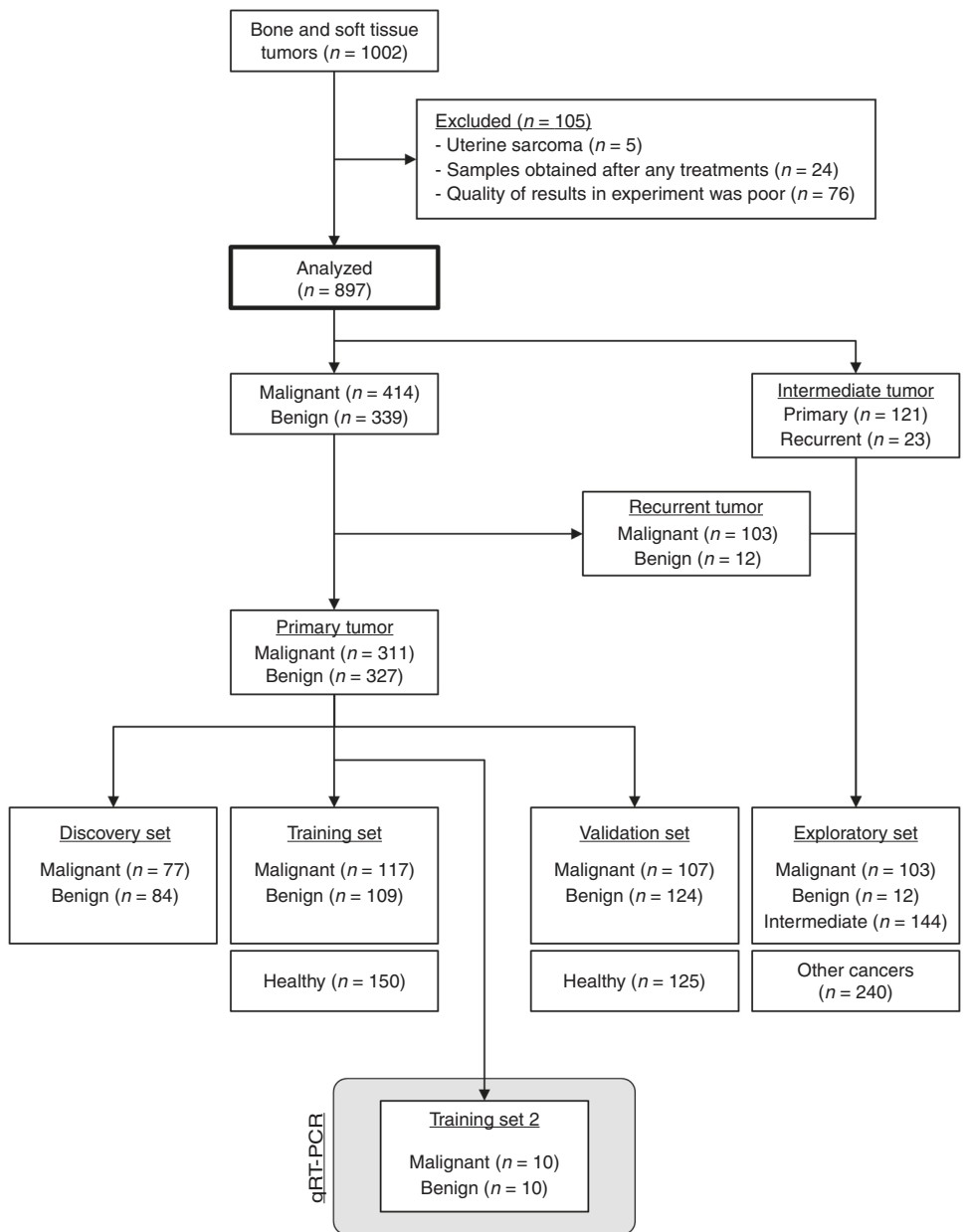

**Fig. 1** Study design. Patients with primary benign or malignant (sarcoma) tumors were divided into discovery, training, and validation cohorts. Training set 2 was used for quantitative RT-PCR analysis. Patients with intermediate, recurrent benign, and recurrent malignant tumors were assigned to the exploratory cohort

the other hand, we identified no clusters in which expression levels decreased in a similar step-by-step manner (Supplementary Figure 2).

Subsequently, we performed qRT-PCR analysis of ten malignant and ten benign samples. Among the 33 miRNAs, miR-4488, miR-1908-3p, miR-1292-3p, miR-6088, and miR-4492 were excluded because specific primers for these sequences could not be obtained. According to the results of qRT-PCR of the remaining 28 miRNAs, we selected 12 miRNAs whose expression levels were higher in malignant samples than in benign samples ($P < 0.05$) (Supplementary Figure 3).

We then analyzed the ability of these 12 miRNAs to detect malignant bone and soft tissue tumors (sarcoma). The results, including sensitivity, specificity, positive predictive value (PPV), negative predictive value (NPV), and area under the receiver operating characteristics (ROC) curve (AUC) are shown in the upper row of Table 2. Although miR-4736 exhibited the best

performance out of the 12 miRNAs, as indicated by the AUC value (0.92; 95% CI, 0.89–0.95), its specificity in the validation cohort was relatively low (0.83; 95% CI, 0.78–0.88). These results suggested that it is difficult to accurately detect malignant sarcoma patients using the level of a single miRNA. Unsupervised cluster analysis and PCA revealed that the serum levels of the 12 selected miRNAs could clearly separate malignant tumors, benign tumors, and healthy controls (Fig. 2c, d). Therefore, we calculated the index in the training cohort using combinations of miRNAs. The best indices using combinations of 2−9 miRNAs, obtained using cross-validation in the training cohort, are shown in the lower row of Table 2. For example, Index I was calculated using two miRNAs with the following formula: $(0.90 \times \text{miR-4736}) + (0.55 \times \text{miR-6836-3p}) - 11.3$. Index II included three miRNAs: $(0.89 \times \text{miR-4736}) + (0.54 \times \text{miR-6836-3p}) + (0.043 \times \text{miR-4281}) - 11.6$. In the training set, combinations of miRNAs improved their performance, and a combination of seven

**Table 1 Participant characteristics**

|  | Discovery set | Training set | Training set 2 | Validation set | P value[a] |
|---|---|---|---|---|---|
| **Malignant** | n = 77 | n = 117 | n = 10 | n = 107 |  |
| Age (y) | 45.9 ± 23.6 | 50.1 ± 21.7 | 47.7 ± 29.7 | 43.8 ± 22.9 | 0.22 |
| Sex |  |  |  |  |  |
| Men | 46 (59.7%) | 70 (59.8%) | 4 (40.0%) | 69 (64.5%) | 0.47 |
| Women | 31 (40.3%) | 47 (40.2%) | 6 (60.0%) | 38 (35.5%) |  |
| Primary site |  |  |  |  |  |
| Head and neck | 5 (6.5%) | 8 (6.8%) | 1 (10.0%) | 6 (5.6%) | 0.98 |
| Body trunk | 33 (42.9%) | 57 (48.7%) | 4 (40.0%) | 50 (46.7%) |  |
| Upper extremities | 5 (6.5%) | 6 (5.1%) | 0 (0%) | 8 (7.5%) |  |
| Lower extremities | 34 (44.2%) | 46 (39.3%) | 5 (50.0%) | 43 (40.2%) |  |
| Bone | n = 20 (26.0%) | n = 23 (19.7%) | n = 3 (30.0%) | n = 32 (29.9%) |  |
| T |  |  |  |  |  |
| T1 | 3 (15.0%) | 12 (52.2%) | 0 (0%) | 17 (53.1%) | 0.044 |
| T2 | 16 (80.0%) | 11 (47.8%) | 3 (100%) | 15 (46.9%) |  |
| T3 | 1 (5.0%) | 0 (0%) | 0 (0%) | 0 (0%) |  |
| N |  |  |  |  |  |
| N0 | 20 (100%) | 23 (100%) | 3 (100%) | 31 (96.9%) | 0.69 |
| N1 | 0 (0%) | 0 (0%) | 0 (0%) | 1 (3.1%) |  |
| M |  |  |  |  |  |
| M0 | 17 (85.0%) | 21 (91.3%) | 2 (66.7%) | 26 (81.3%) | 0.17 |
| M1a | 1 (5.0%) | 2 (8.7%) | 0 (0%) | 5 (15.6%) |  |
| M1b | 2 (10.0%) | 0 (0%) | 1 (33.3%) | 1 (3.1%) |  |
| G |  |  |  |  |  |
| Low | 2 (10.0%) | 0 (0%) | 0 (0%) | 0 (0%) | 0.17 |
| High | 17 (85.0%) | 23 (100%) | 3 (100%) | 32 (100%) |  |
| GX | 1 (5.0%) | 0 (0%) | 0 (0%) | 0 (0%) |  |
| Stage |  |  |  |  |  |
| IA | 1 (5.0%) | 0 (0%) | 0 (0%) | 0 (0%) | 0.15 |
| IB | 1 (5.0%) | 0 (0%) | 0 (0%) | 0 (0%) |  |
| IIA | 2 (10.0%) | 12 (52.2%) | 0 (0%) | 17 (53.1%) |  |
| IIB | 12 (60.0%) | 9 (39.1%) | 2 (66.7%) | 7 (21.9%) |  |
| III | 1 (5.0%) | 0 (0%) | 0 (0%) | 2 (6.3%) |  |
| IVA | 1 (5.0%) | 1 (4.3%) | 0 (0%) | 3 (9.4%) |  |
| IVB | 2 (10.0%) | 1 (4.3%) | 1 (33.3%) | 3 (9.4%) |  |
| Soft tissue | n = 57 (74.0%) | n = 94 (80.3%) | n = 7 (70.0%) | n = 75 (70.1%) |  |
| T |  |  |  |  |  |
| T1a | 5 (8.8%) | 8 (8.5%) | 2 (28.6%) | 5 (6.7%) | 0.48 |
| T1b | 6 (10.5%) | 11 (11.7%) | 2 (28.6%) | 7 (9.3%) |  |
| T2a | 2 (3.5%) | 7 (7.4%) | 0 (0%) | 7 (9.3%) |  |
| T2b | 44 (77.2%) | 66 (70.2%) | 3 (42.9%) | 53 (70.7%) |  |
| TX | 0 (0%) | 2 (2.1%) | 0 (0%) | 3 (4.0%) |  |
| N |  |  |  |  |  |
| N0 | 54 (94.7%) | 82 (87.2%) | 7 (100%) | 69 (92.0%) | 0.34 |
| N1 | 3 (5.3%) | 12 (12.8%) | 0 (0%) | 6 (8.0%) |  |
| M |  |  |  |  |  |
| M0 | 44 (77.2%) | 80 (85.1%) | 6 (85.7%) | 61 (81.3%) | 0.80 |
| M1a | 9 (15.8%) | 12 (12.8%) | 1 (14.3%) | 10 (13.3%) |  |
| M1b | 4 (7.0%) | 2 (1.7%) | 0 (0%) | 4 (5.3%) |  |
| G |  |  |  |  |  |
| Low | 8 (14.0%) | 12 (12.8%) | 1 (14.3%) | 10 (13.3%) | 1.00 |
| High | 49 (86.0%) | 82 (87.2%) | 6 (85.7%) | 65 (86.7%) |  |
| Stage |  |  |  |  |  |
| IA | 0 (0%) | 2 (2.1%) | 0 (0%) | 2 (2.7%) | 0.42 |
| IB | 7 (12.3%) | 9 (9.6%) | 1 (14.3%) | 6 (8.0%) |  |
| IIA | 10 (17.5%) | 13 (13.8%) | 4 (57.1%) | 9 (12.0%) |  |
| IIB | 1 (1.8%) | 8 (8.5%) | 0 (0%) | 5 (6.7%) |  |
| III | 26 (45.6%) | 44 (46.8%) | 1 (14.3%) | 37 (49.3%) |  |
| IV | 13 (22.8%) | 17 (18.1%) | 1 (14.3%) | 14 (18.7%) |  |
| X | 0 (0%) | 1 (1.1%) | 0 (0%) | 2 (2.7%) |  |
| **Benign** | n = 84 | n = 109 | n = 10 | n = 124 |  |
| Age (y) | 44.8 ± 18.4 | 43.8 ± 18.4 | 41.0 ± 22.9 | 44.5 ± 19.1 | 0.93 |
| Sex |  |  |  |  |  |
| Men | 42 (50.0%) | 58 (53.2%) | 6 (60.0%) | 52 (41.9%) | 0.30 |
| Women | 42 (50.0%) | 51 (46.8%) | 4 (40.0%) | 72 (58.1%) |  |
| Primary site[b] |  |  |  |  |  |
| Head and neck | 4 (4.7%) | 2 (1.8%) | 0 (0%) | 6 (4.8%) | 0.88 |
| Body trunk | 26 (30.6%) | 29 (25.7%) | 2 (20.0%) | 40 (32.0%) |  |
| Upper extremities | 21 (24.7%) | 31 (27.4%) | 3 (30.0%) | 31 (24.8%) |  |
| Lower extremities | 34 (40.0%) | 51 (45.1%) | 5 (50.0%) | 48 (38.4%) |  |
| **Healthy** | n = 0 | n = 150 | n = 0 | n = 125 |  |
| Age (y) |  | 51.2 ± 12.2 |  | 51.1 ± 12.1 | 0.96 |
| Sex |  |  |  |  |  |
| Men |  | 82 (54.7%) |  | 68 (54.4%) | 0.97 |
| Women |  | 68 (45.3%) |  | 57 (45.6%) |  |

[a]Comparisons among four groups were performed using Pearson's $\chi^2$ test for categorical variables, Student's $t$ test for two continuous variables, and one-way ANOVA for three continuous variables
[b]Six patients had tumors at multiple sites

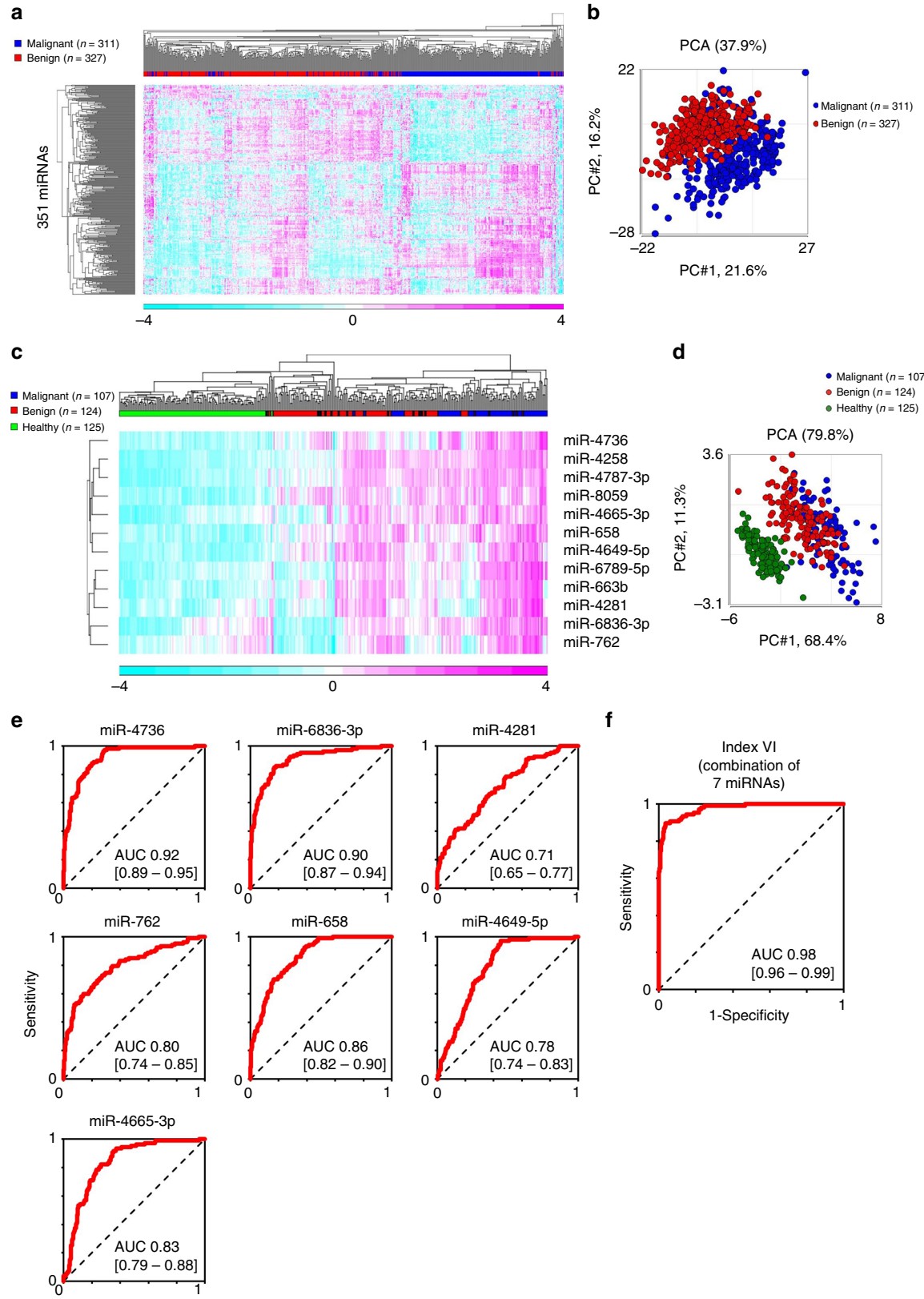

**Fig. 2** Serum miRNA profiles in patients with bone and soft tissue tumors. Serum miRNA profiles were compared between malignant (sarcoma) and benign tumors by **a** unsupervised cluster analysis, and **b** principal component analysis (PCA). Profiles and performance of 12 selected miRNAs to detect sarcomas were analyzed by **c** unsupervised cluster analysis and **d** PCA. **e** Receiver operating characteristics (ROC) analyses of the performance of the seven miRNAs included in Index VI in the validation set. **f** ROC analyses of the performance of Index VI in the validation set

**Table 2 Predictive accuracy in the training and validation sets**

| | Training set | | | | | | Validation set | | | | | |
|---|---|---|---|---|---|---|---|---|---|---|---|---|
| | Sensitivity (95% CI) | Specificity (95% CI) | PPV (95% CI) | NPV (95% CI) | Accuracy (95% CI) | AUC (95% CI) | Sensitivity (95% CI) | Specificity (95% CI) | PPV (95% CI) | NPV (95% CI) | Accuracy (95% CI) | AUC (95% CI) |
| miR-4736 | 0.88 (0.82–0.94) | 0.88 (0.84–0.92) | 0.77 (0.70–0.85) | 0.94 (0.91–0.97) | 0.88 (0.85–0.92) | 0.95 (0.92–0.97) | 0.81 (0.74–0.89) | 0.83 (0.78–0.88) | 0.67 (0.59–0.76) | 0.91 (0.87–0.95) | 0.83 (0.79–0.87) | 0.92 (0.89–0.95) |
| miR-6836-3p | 0.90 (0.84–0.95) | 0.78 (0.73–0.83) | 0.65 (0.57–0.72) | 0.94 (0.91–0.97) | 0.82 (0.78–0.86) | 0.91 (0.89–0.94) | 0.93 (0.87–0.98) | 0.71 (0.65–0.76) | 0.58 (0.50–0.65) | 0.96 (0.93–0.99) | 0.77 (0.73–0.82) | 0.90 (0.87–0.94) |
| miR-4258 | 0.91 (0.86–0.97) | 0.78 (0.73–0.83) | 0.66 (0.58–0.73) | 0.95 (0.92–0.98) | 0.82 (0.79–0.86) | 0.90 (0.87–0.93) | 0.93 (0.89–0.98) | 0.71 (0.65–0.77) | 0.58 (0.51–0.66) | 0.96 (0.93–0.99) | 0.78 (0.73–0.82) | 0.86 (0.83–0.90) |
| miR-4787-3p | 0.91 (0.86–0.97) | 0.78 (0.73–0.83) | 0.66 (0.58–0.73) | 0.95 (0.92–0.98) | 0.82 (0.79–0.86) | 0.91 (0.88–0.94) | 0.87 (0.80–0.93) | 0.69 (0.63–0.75) | 0.55 (0.47–0.62) | 0.92 (0.89–0.96) | 0.74 (0.70–0.79) | 0.87 (0.83–0.91) |
| miR-6789-5p | 0.90 (0.84–0.95) | 0.75 (0.69–0.80) | 0.61 (0.54–0.69) | 0.94 (0.91–0.97) | 0.79 (0.75–0.83) | 0.88 (0.85–0.92) | 0.93 (0.87–0.98) | 0.71 (0.65–0.76) | 0.58 (0.50–0.65) | 0.96 (0.93–0.99) | 0.77 (0.73–0.82) | 0.89 (0.86–0.93) |
| miR-658 | 0.97 (0.93–1.00) | 0.66 (0.60–0.72) | 0.56 (0.49–0.63) | 0.98 (0.95–1.00) | 0.76 (0.71–0.80) | 0.88 (0.85–0.91) | 0.93 (0.89–0.98) | 0.59 (0.53–0.66) | 0.50 (0.43–0.57) | 0.95 (0.92–0.99) | 0.70 (0.65–0.74) | 0.86 (0.82–0.90) |
| miR-8059 | 0.92 (0.87–0.97) | 0.67 (0.61–0.73) | 0.56 (0.49–0.63) | 0.95 (0.92–0.98) | 0.75 (0.70–0.79) | 0.84 (0.80–0.88) | 0.91 (0.85–0.96) | 0.57 (0.51–0.63) | 0.48 (0.41–0.54) | 0.93 (0.89–0.97) | 0.67 (0.62–0.72) | 0.80 (0.75–0.84) |
| miR-4665-3p | 0.86 (0.80–0.93) | 0.75 (0.70–0.80) | 0.61 (0.53–0.68) | 0.92 (0.89–0.96) | 0.78 (0.74–0.83) | 0.85 (0.81–0.89) | 0.88 (0.82–0.94) | 0.66 (0.60–0.72) | 0.53 (0.45–0.60) | 0.93 (0.89–0.97) | 0.73 (0.68–0.77) | 0.83 (0.79–0.88) |
| miR-663b | 0.73 (0.64–0.81) | 0.75 (0.70–0.80) | 0.57 (0.49–0.65) | 0.86 (0.81–0.90) | 0.74 (0.70–0.79) | 0.79 (0.74–0.84) | 0.75 (0.66–0.83) | 0.71 (0.66–0.77) | 0.53 (0.45–0.61) | 0.87 (0.82–0.91) | 0.72 (0.68–0.77) | 0.80 (0.75–0.85) |
| miR-4649-5p | 0.90 (0.84–0.95) | 0.60 (0.54–0.66) | 0.50 (0.44–0.57) | 0.93 (0.89–0.97) | 0.69 (0.65–0.74) | 0.78 (0.74–0.83) | 0.97 (0.94–1.00) | 0.55 (0.49–0.61) | 0.48 (0.41–0.55) | 0.98 (0.95–1.00) | 0.68 (0.63–0.73) | 0.78 (0.74–0.83) |
| miR-4281 | 0.60 (0.51–0.69) | 0.76 (0.71–0.81) | 0.53 (0.44–0.62) | 0.81 (0.76–0.86) | 0.71 (0.66–0.76) | 0.71 (0.66–0.77) | 0.58 (0.48–0.67) | 0.69 (0.64–0.75) | 0.45 (0.37–0.53) | 0.79 (0.74–0.85) | 0.66 (0.61–0.71) | 0.71 (0.65–0.77) |
| miR-762 | 0.68 (0.60–0.77) | 0.74 (0.69–0.79) | 0.54 (0.46–0.63) | 0.84 (0.79–0.89) | 0.72 (0.68–0.77) | 0.76 (0.71–0.81) | 0.73 (0.64–0.81) | 0.71 (0.66–0.77) | 0.52 (0.44–0.60) | 0.86 (0.81–0.91) | 0.72 (0.67–0.77) | 0.79 (0.74–0.85) |
| Index I[a] | 0.91 (0.86–0.97) | 0.90 (0.87–0.94) | 0.81 (0.74–0.88) | 0.96 (0.93–0.98) | 0.91 (0.88–0.94) | 0.97 (0.95–0.98) | 0.84 (0.77–0.91) | 0.88 (0.84–0.92) | 0.76 (0.68–0.83) | 0.93 (0.90–0.96) | 0.87 (0.84–0.91) | 0.95 (0.93–0.97) |
| Index II[b] | 0.91 (0.86–0.97) | 0.90 (0.86–0.93) | 0.80 (0.73–0.87) | 0.96 (0.93–0.98) | 0.90 (0.87–0.93) | 0.96 (0.95–0.98) | 0.86 (0.79–0.93) | 0.88 (0.83–0.92) | 0.75 (0.67–0.83) | 0.94 (0.90–0.97) | 0.87 (0.84–0.91) | 0.95 (0.93–0.97) |
| Index III[c] | 0.97 (0.95–1.00) | 0.88 (0.84–0.92) | 0.79 (0.72–0.86) | 0.99 (0.97–1.00) | 0.91 (0.88–0.94) | 0.98 (0.97–0.99) | 0.92 (0.86–0.97) | 0.87 (0.83–0.91) | 0.75 (0.68–0.83) | 0.96 (0.93–0.99) | 0.88 (0.85–0.92) | 0.97 (0.95–0.98) |
| Index IV[d] | 0.98 (0.96–1.00) | 0.91 (0.88–0.95) | 0.83 (0.77–0.90) | 0.99 (0.98–1.00) | 0.93 (0.91–0.96) | 0.98 (0.97–0.99) | 0.92 (0.86–0.97) | 0.90 (0.86–0.93) | 0.79 (0.72–0.86) | 0.96 (0.94–0.99) | 0.90 (0.87–0.93) | 0.97 (0.96–0.99) |
| Index V[e] | 0.97 (0.95–1.00) | 0.92 (0.88–0.95) | 0.84 (0.78–0.90) | 0.99 (0.97–1.00) | 0.93 (0.91–0.96) | 0.98 (0.98–0.99) | 0.91 (0.85–0.96) | 0.90 (0.86–0.94) | 0.80 (0.72–0.87) | 0.96 (0.93–0.98) | 0.90 (0.87–0.93) | 0.98 (0.96–0.99) |
| Index VI[f] | 0.96 (0.92–0.99) | 0.95 (0.92–0.97) | 0.89 (0.83–0.94) | 0.98 (0.96–1.00) | 0.95 (0.93–0.97) | 0.99 (0.98–0.99) | 0.90 (0.84–0.96) | 0.95 (0.92–0.98) | 0.88 (0.82–0.94) | 0.96 (0.93–0.98) | 0.93 (0.91–0.96) | 0.98 (0.96–0.99) |
| Index VII[g] | 0.96 (0.92–0.99) | 0.95 (0.92–0.97) | 0.89 (0.83–0.94) | 0.98 (0.96–1.00) | 0.95 (0.93–0.97) | 0.99 (0.98–0.99) | 0.90 (0.84–0.96) | 0.95 (0.92–0.98) | 0.88 (0.82–0.94) | 0.96 (0.93–0.98) | 0.93 (0.91–0.96) | 0.98 (0.96–0.99) |
| Index VIII[h] | 0.96 (0.92–0.99) | 0.94 (0.91–0.97) | 0.88 (0.82–0.93) | 0.98 (0.96–1.00) | 0.94 (0.92–0.97) | 0.99 (0.98–0.99) | 0.90 (0.84–0.96) | 0.94 (0.91–0.97) | 0.87 (0.81–0.94) | 0.96 (0.93–0.98) | 0.93 (0.90–0.96) | 0.98 (0.96–0.99) |

*CI* confidence interval, *PPV* positive predictive value, *NPV* negative predictive value, *AUC* area under the receiver operating characteristics (ROC) curve

[a]Calculated as (0.90 × miR-4736) + (0.55 × miR-6836-3p) − 11.3
[b]Calculated as (0.89 × miR-4736) + (0.54 × miR-6836-3p) + (0.043 × miR-4281) − 11.6
[c]Calculated as (1.00 × miR-4736) + (0.58 × miR-6836-3p) − (1.01 × miR-4281) + (0.91 × miR-762) − 12.5
[d]Calculated as (0.84 × miR-4736) + (0.49 × miR-6836-3p) − (1.30 × miR-4281) + (1.26 × miR-762) + (0.43 × miR-658) − 14.8
[e]Calculated as (0.86 × miR-4736) + (0.45 × miR-6836-3p) − (1.13 × miR-4281) + (1.34 × miR-762) + (0.58 × miR-658) − (0.25 × miR-4649-5p) − 16.0
[f]Calculated as (0.87 × miR-4736) + (0.52 × miR-6836-3p) − (1.14 × miR-4281) + (1.31 × miR-762) + (0.59 × miR-658) − (0.22 × miR-4649-5p) − (0.15 × miR-4665-3p) − 15.9
[g]Calculated as (0.87 × miR-4736) + (0.54 × miR-6836-3p) − (1.12 × miR-4281) + (1.28 × miR-762) + (0.59 × miR-658) − (0.20 × miR-4649-5p) − (0.15 × miR-4665-3p) − (0.035 × miR-663b) − 15.8
[h]Calculated as (0.87 × miR-4736) + (0.55 × miR-6836-3p) − (1.11 × miR-4281) + (1.29 × miR-762) + (0.60 × miR-658) − (0.18 × miR-4649-5p) − (0.14 × miR-4665-3p) − (0.038 × miR-663b) − (0.034 × miR-4258) − 16.1

miRNAs (Index VI) reached the highest accuracy [sensitivity of 0.96 (95% CI, 0.92–0.99), specificity of 0.95 (95% CI, 0.92–0.97), PPV of 0.89 (95% CI, 0.83–0.94), NPV of 0.98 (95% CI, 0.96–1.00), accuracy of 0.95 (95% CI, 0.93–0.97), and AUC of 0.99 (95% CI, 0.98–0.99)] (Table 2). No eight- or nine-miRNA combination increased discrimination accuracy greater than that of the seven-miRNA combination (Index VI). The diagnostic performance of Index VI was confirmed in the validation set [sensitivity of 0.90 (95% CI, 0.84–0.96), specificity of 0.95 (95% CI, 0.92–0.98), PPV of 0.88 (95% CI, 0.82–0.94), NPV of 0.96 (95% CI, 0.93–0.98), accuracy of 0.93 (95% CI, 0.91–0.96), and AUC of 0.98 (95% CI, 0.96–0.99)] (Table 2). The ability of Index VI to detect sarcoma was preserved after adjustment for age and sex (Supplementary Table 5). Therefore, we chose Index VI for further validation as a biomarker for sarcomas. The ROC curves for Index VI and each of its seven-component miRNAs are shown in Fig. 2e, f.

**The utility of Index VI as a biomarker.** Although Index VI could clearly discriminate malignant tumors from benign tumors or healthy controls, intermediate tumors were divided into two populations, the malignant group (38.0%) and the benign group (62.0%) (Fig. 3a). Hence, we further evaluated the utility of Index VI in distinguishing the main histological subtypes of sarcoma. In bone sarcomas, the discriminant accuracy of Index VI remained

high, and the values did not differ significantly among the histological subtypes examined (malignant bone tumor: $P = 0.02$). A similar trend was observed in malignant soft tissue tumors ($P = 0.14$, Fig. 3b). Among benign tumors, the discriminant accuracy of Index VI remained low, and the values did not differ significantly among the histological subtypes (benign bone tumor: $P = 0.20$; and benign soft tissue tumor: $P = 0.49$, Fig. 3d). Index VI values for three subtypes of intermediate tumors, including atypical cartilaginous tumor/chondrosarcoma grade I (CS grade I) ($P < 0.001$, Fig. 3c left), atypical lipomatous tumor/well-differentiated liposarcoma (WDLPS), and dermatofibrosarcoma protuberans (DFSP) ($P < 0.001$, Fig. 3c right), were very close to those for sarcomas, suggesting that serum miRNA profiles might reflect the premalignant potential of these subtypes.

Next, we assessed whether other clinical factors affect the Index VI value. As indicated in Supplementary Figures 4 and 5, TNM stage[17] (malignant bone tumor: $P = 0.32$, Supplementary Figure 4A; malignant soft tissue tumor: $P = 0.99$, Supplementary Figure 4B), tumor origin (malignant: $P = 0.23$; benign: $P = 0.14$; Supplementary Figure 5A), tumor location (malignant: $P = 0.94$; benign: $P = 0.95$; Supplementary Figure 5B), and tumor status (primary vs. recurrent) (intermediate: $P = 0.47$; benign: $P = 0.22$; Supplementary Figure 5C) were not significantly associated with Index VI values. Only recurrent cases of malignant tumors had considerable numbers of low-Index VI values (malignant: $P < 0.001$, Supplementary Figure 5C). Taken together, these results

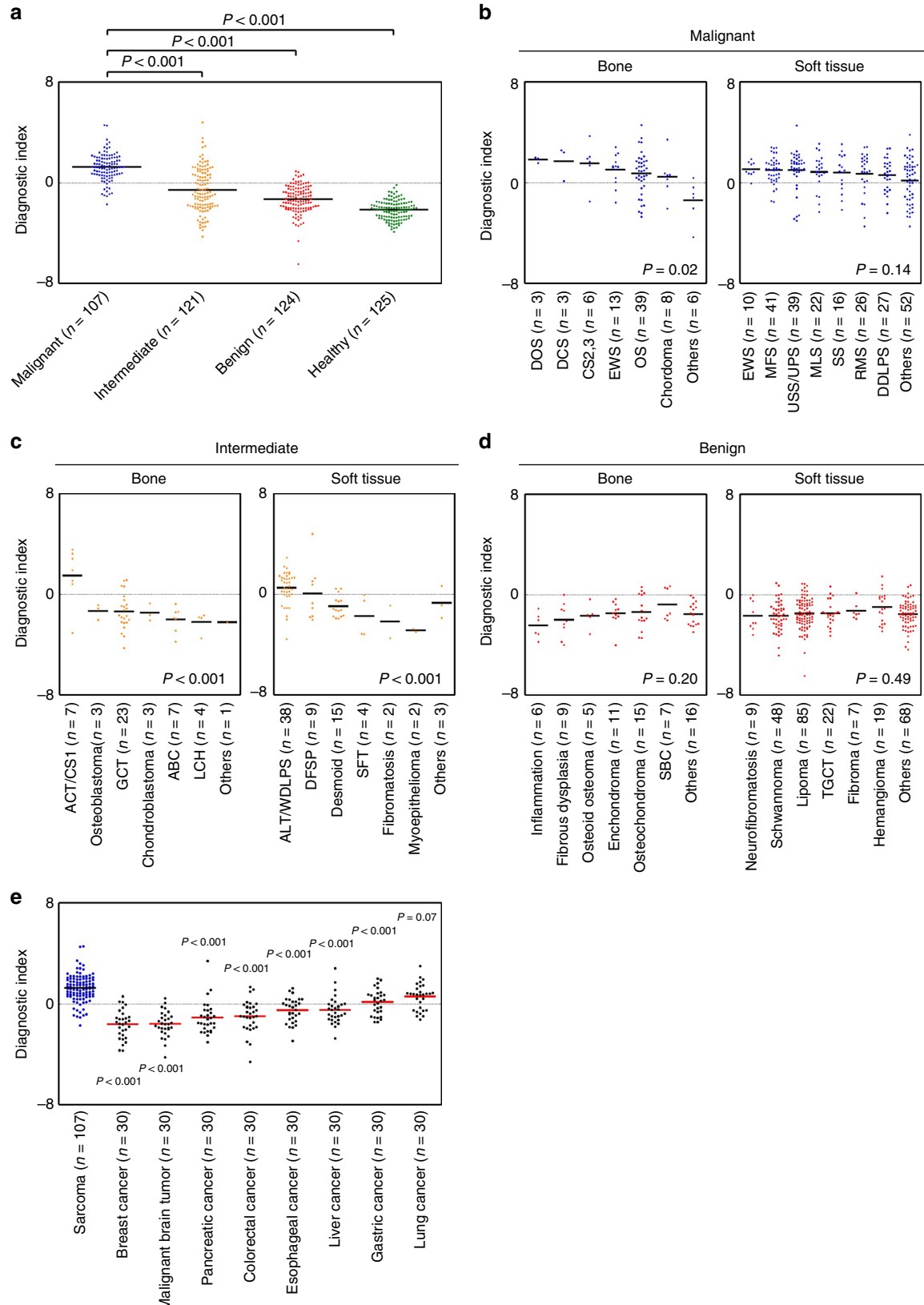

suggest that Index VI represents a promising classifier for the detection of sarcomas occurring at any anatomic site, even in the early stages. To further assess whether Index VI is specific to sarcomas, we performed a comparative analysis of diagnostic power using serum from patients with other types of cancer, including lung, breast, esophageal, gastric, colorectal hepatocellular, pancreatic, and brain tumors. Sarcomas yielded the highest Index VI values among all cancer types tested, with the exception of lung cancer ($P = 0.07$, Fig. 3e), indicating that Index VI is a powerful tool for the detection of sarcomas.

**Fig. 3** Distinct patterns of Index VI in bone and soft tissue tumors. **a** Values of Index VI differed significantly between malignant (sarcoma) tumors and the other samples in the validation and exploratory sets. The Index VI values of the discovery, training, and validation or exploratory sets were compared among various histological subtypes in **b** malignant (sarcoma), **c** intermediate, and **d** benign tumors. **e** The values of Index VI were also compared between sarcomas and the other eight cancers in the validation and exploratory sets. CS2,3 chondrosarcoma, grades II and III, DOS differentiated osteosarcoma, DCS differentiated chondrosarcoma, EWS Ewing sarcoma, OS osteosarcoma, MLS myxoid liposarcoma, MFS myxofibrosarcoma, USS/UPS undifferentiated spindle cell sarcoma/undifferentiated pleomorphic sarcoma, RMS rhabdomyosarcoma, SS synovial sarcoma, DDLPS dedifferentiated liposarcoma, ACT/CS1 atypical cartilaginous tumor/chondrosarcoma, grade I, ABC aneurysmal bone cyst, GCT giant cell tumor of bone, LCH Langerhans cell histiocytosis, ATL/WDLPS atypical lipomatous tumor/well-differentiated liposarcoma, DFSP dermatofibrosarcoma protuberans, Desmoid desmoid-type fibromatosis, SFT solitary fibrous tumor, SBC simple bone cyst, TGCT tenosynovial giant cell tumor

**Expression of seven serum miRNAs in sarcoma tissues**. To determine whether the seven serum miRNAs that constitute Index VI originated from tumor tissues, we compared their expression levels between normal tissues (bone, fat, and muscle) and sarcoma tissues (osteosarcoma and dedifferentiated liposarcoma). Although the serum levels of all seven miRNAs were higher in malignant samples, none were significantly upregulated in sarcoma (Supplementary Figure 6A). In addition, no serum miRNAs were correlated with those in tissues from sarcoma patients (Supplementary Figure 6B). These results indicate that the seven circulating miRNAs may have been reactively upregulated by tumor-associated conditions such as inflammatory reactions or immune activity.

**Discussion**

In this study, we performed comprehensive serum miRNA profiling of approximately 1000 samples of bone and soft tissue tumors belonging to 43 histological subtypes, and identified a promising classifier for the detection of malignant cases, namely, sarcomas. This cohort is the largest analyzed to date, and our approach resolved an important problem that has limited the development of diagnostic strategies for the detection of sarcomas. On the basis of these data, we identified a combination of seven miRNAs capable of discriminating cases of sarcoma. In addition, we showed that a subset of miRNAs is dysregulated in the presence of sarcoma cells regardless of histological subtype and primary tumor site.

Circulating miRNAs in blood have recently attracted attention as potential diagnostic biomarkers for cancer detection. Their potential utility is based on an important feature of circulating miRNAs, namely, their cancer origin-specific expression signature[15]. In this study, bone and soft tissue sarcomas exhibited similar circulating miRNA profiles despite the inclusion of various histological subtypes in our cohort. Index VI, which is based on the levels of seven miRNAs in sera, exhibited the best performance for the classification of sarcomas vs. benign tumors or healthy controls, indicating that it represents a promising classifier for sarcomas, and further suggesting that circulating miRNAs reflect the mesenchymal origin of sarcomas. This property was not observed in previous analyses using small sample sizes or limited subtypes of sarcomas, and it is possible that its identification was only possible in a large comprehensive analysis using a standardized platform. Several of the previous reports that comprehensively analyzed serum miRNAs with the goal of identifying diagnostic biomarkers used only a few serum samples from patients: one cohort consisted of 20 samples of malignant peripheral nerve sheath tumors and 10 neurofibromas[18]; another used material from five synovial sarcomas[19]; and another used three rhabdomyosarcomas, two Ewing sarcomas, and one osteosarcoma[20]. Although each of these studies identified several serum miRNAs as potential diagnostic biomarkers, the candidate miRNAs could not be sufficiently validated. As expected from our larger sample size, we identified serum miRNAs that were not

reported in these studies, likely reflecting the difficulties associated with distinguishing the important features of sarcomas among histological subtypes in individual analyses.

Index VI, which exhibited the ability to diagnose sarcoma accurately, was calculated based on the levels of seven serum miRNAs: miR-4736, miR-6836-3p, miR-4281, miR-762, miR-658, miR-4649-5p, and miR-4665-3p. We confirmed the reproducibility of the selected miRNAs in a smaller sample set using a qRT-PCR platform. The diagnostic power of Index VI was remarkably higher than that of 12 individual miRNAs whose serum expression levels differed significantly between sarcomas and benign tumors (Table 2), indicating that serum miRNA profiles reflect cancer cell properties or changes in reactivity induced by cancer cell development. From this standpoint, the use of combinations of miRNAs represents the optimal diagnostic strategy.

Index VI was not affected by other clinical factors such as histological subtype, primary site, or stage. In contrast to conventional tumor markers such as PSA and CEA, which are specifically released from tumor cells, Index VI did not increase with tumor burden. In addition, the levels of the seven-component miRNAs were not correlated between tumor tissue and serum (Supplementary Figure 6). Together, these observations suggest that, unlike tumor-released conventional biomarkers, these miRNA biomarkers are not secreted from tumors. These features do not necessarily decrease the utility of Index VI, because its strong performance in discriminating benign from malignant cases does not require it to increase with tumor burden.

Although Index VI effectively discriminated sarcomas from benign tumors or healthy controls, intermediate tumors contained high- and low-Index VI populations (Fig. 3a). Analysis of Index VI in individual subtypes of intermediate tumors revealed that only three subtypes, CS grade I, WDLPS, and DFSP, had higher Index VI values than other histological subtypes. Approximately 10–15% of tumors in these three subtypes can undergo transformation to higher histologic phenotypes (CS grade II, dedifferentiated liposarcoma, and fibrosarcomatous DFSP, respectively)[1]. This indicates that populations of intermediate tumors with high-Index VI values in this cohort included a considerable number of tumors with malignant transformation potential. The data from intermediate tumors indicated that serum levels of seven-component miRNAs of Index VI reflect common biological features of potentially malignant phenotypes in sarcoma cells, and these properties can be acquired in the early stages of sarcoma development.

Index VI had the highest diagnostic power for detection of sarcoma among eight types of cancer; however, several types of cancer, especially lung cancer, included a considerable number of cases with Index VI values similar to those of sarcoma (Fig. 3e). These results suggest that the serum levels of the seven-component miRNAs reflect the reactivity change induced by cancer-associated conditions in the different tumor types.

Serum miRNA profiles mirror the specific changes that occur in cancer patients, including inflammation and altered immune responses. We exploited this feature to develop Index VI, which

had high diagnostic value for detection of early-stage sarcomas (Supplementary Figure 4A and B). The seven-component miR-NAs of Index VI are probably secreted by various types of tumor-associated cells. Identification of the cells that secrete sarcoma-specific miRNAs represents a critical challenge for future research. Moreover, the biological functions of serum miRNAs associated with sarcoma development and progression are currently unclear. Accordingly, future research should also seek to determine how these miRNAs contribute to cancer development, including tumor cell adaptation to microenvironments.

Index VI had a remarkable ability to diagnose sarcomas, but could not distinguish between histological subtypes. Patients with certain histological subtypes, such as osteosarcoma, Ewing sarcoma, and rhabdomyosarcoma, undergo neoadjuvant chemotherapy as a standard treatment. To detect specific histological subtypes of sarcomas in a clinical context, more accurate diagnostic methods will be required. The biological properties of extracellular miRNAs could be used to identify serum miRNAs and/or miRNA combinations for the detection of specific histological subtypes of sarcomas. However, because these tumors are rare, this prospect is hampered by the limited availability of serum samples from patients with individual histological subtypes. Therefore, we propose that, for the time being, Index VI should be restricted to the differential diagnosis of benign and malignant bone and soft tissue tumors, and/or differential diagnosis of sarcoma and other cancers. In particular, the method could facilitate differential diagnosis for cases in which diagnostic complications occur. Furthermore, Index VI could be used to screen patients with sarcomas during medical examinations. To implement the clinical use of Index VI, further analyses will be required. A prospective study is currently underway to determine whether Index VI can be used as a classifier for sarcomas.

In summary, comprehensive analysis of serum miRNA profiles in approximately 1000 cases of bone and soft tissue tumors identified a promising classifier, Index VI, which was calculated using the serum levels of seven miRNAs. This classifier exhibited remarkable performance for the detection of sarcoma. One of the most important issues in cancer medicine is the accurate detection of malignant tumors at an early stage, which improves the prognosis of patients. From this standpoint, the development of noninvasive, rapid, and accurate diagnostic methods for sarcoma is an important challenge that must be overcome. The present findings overcome a serious problem associated with sarcoma diagnosis and provide the basis for the development and implementation of miRNA-based strategies for clinical diagnosis.

## Methods

**Study design and participants**. The National Cancer Center Hospital (NCCH) contains a rare cancer center, and many patients with rare cancers such as sarcoma were recruited from all over Japan. Serum samples were obtained consecutively from 1002 patients with bone and soft tissue tumors who were admitted or referred to the NCCH between 2007 and 2013. All serum samples were frozen at −20 °C within 12 h of collection using serum collection tubes (VP-AS109K50 or VP-AR109K63, TERUMO, Tokyo, Japan). Patients with uterine sarcoma and those treated with chemotherapy or radiotherapy before serum collection were excluded, as were patients with poor-quality microarray data.

Control serum samples were obtained from healthy volunteers aged >35 years who were recruited from the Yokohama Minoru Clinic in 2015. The inclusion criteria for this cohort were no history of cancer and no hospitalization during the previous 3 months. Serum samples were stored at −80 °C before use.

Serum samples were obtained from patients with eight types of cancer (breast, brain, pancreatic, colorectal, hepatic, esophageal, gastric, and lung cancers; n = 30 for each) randomly selected from among patients admitted or referred to the NCCH between 2008 and 2013 and registered in the National Cancer Center Biobank. Serum samples were stored at −20 °C until use.

Frozen tumor tissue samples of osteosarcoma and dedifferentiated liposarcoma were obtained from patients who underwent surgery at the NCCH between 2007 and 2013. Normal bone, fat, and muscle samples were also obtained from patients who underwent surgery at the NCCH in 2016. All tissue samples were cryopreserved in liquid nitrogen until use.

This study was approved by the NCCH Institutional Review Board (2004-050, 2013-111, 2015-266) and the Research Ethics Committee of Medical Corporation Shintokai Yokohama Minoru Clinic (6019-18-3772). Written informed consent was obtained from each participant.

**MiRNA expression array of serum samples**. Total RNA was extracted from 300 μL of serum using the 3D-Gene® RNA extraction reagent (Toray Industries, Inc., Tokyo, Japan). One-half of the amount of extracted total RNA was prepared for comprehensive miRNA expression analysis using the 3D-Gene® miRNA Labeling kit, and then analyzed on the 3D-Gene® Human miRNA Oligo Chip (Toray Industries, Inc.), which was designed to detect 2555 miRNA sequences registered in miRBase release 20 or 2565 miRNA sequences registered in miRBase release 21 (http://www.mirbase.org/).

The microarray data were subjected to quality control according to the following criteria: coefficient of variation for negative control probes of <0.15, and number of flagged probes <10. Flagged probes indicate uneven hybridization, which was automatically identified by the 3D-Gene® Scanner. An miRNA was considered to be present if the corresponding microarray signal was greater than [mean + 2 × standard deviation (SD)] of the negative control signals, from which the top and bottom 5%, ranked by signal intensity, were removed. Once an miRNA was considered present, the mean signal of the negative controls (from which the top and bottom 5%, ranked by signal intensity, were removed) was subtracted from the miRNA signal. When a signal value was negative (or undetected) after background subtraction, the value was replaced by the lowest signal intensity on the microarray minus 0.1 on a base-2 logarithmic scale. To normalize the signals among microarrays, three preselected internal control miRNAs (miR-149-3p, miR-2861, and miR-4463), which had been stably detected in more than 500 serum samples, were used. Each miRNA signal value was standardized using the ratio of the average signal of the three internal control miRNAs to the preset value. All microarray data of this study were in agreement with the Minimum Information About a Microarray Experiment (MIAME) guidelines.

**Quantitative RT-PCR**. Total RNA was extracted using the same methods as for microarray analysis and subjected to quantitative RT-PCR (qRT-PCR) analysis. Ten samples from patients with malignant bone or soft tissue sarcoma and ten samples from patients with benign bone or soft tissue tumors were selected and compared. MiRNA expression levels were investigated using miScript Primer Assays (QIAGEN, Supplementary Table 6) with the miScript II RT kit, miScript PreAMP PCR kit, and miScript SYBR Green PCR kit. The expression levels were normalized against the average signal of miR-149-3p, miR-2861, and miR-4463. The $2^{-\Delta Ct}$ method was used to compare normalized expression levels. Only 28 of the 33 candidate miRNAs were analyzed because miScript PCR primers for miR-4488, miR-1908-3p, miR-1292-3p, miR-6088, and miR-4492 were not commercially available.

**Statistical analysis**. Before statistical comparisons, samples of primary bone and soft tissue tumors were divided into three cohorts: discovery (patients admitted or referred to NCCH between 2011/7 and 2013/12), training, and validation cohorts (between 2007/1 and 2011/6). The training and validation cohorts were randomly and blindly divided. The discovery cohort was used to select significant miRNA markers; the training cohort was used to validate miRNA markers and identify candidates for discriminant models; and the validation cohort was used to validate the discriminant models.

Because the samples were chronologically divided into three cohorts, the sample size for each cohort could not be determined based on a prespecified effect size. To use serum miRNAs as discrimination biomarkers, the required minimum difference between two groups was 0.5 (in log₂ scale). The representative SD was approximately 1 on a log₂ scale, as shown in Supplementary Table 3. Assuming an effect size of 0.5, an SD of 1, an α-error of 0.05, and a power of 0.85, 72 participants would be required for each group. Given that more than 72 samples were enrolled in the benign and malignant groups irrespective of the cohorts, the desired sample size is considered to have been achieved.

Using the discovery cohort, only miRNAs with a signal value $>2^6$ in more than 50% of the analyses in the sarcoma group or benign tumor group were defined as abundant miRNAs in sera. To identify robust biomarkers for discriminating between malignant and benign samples, a cross-validation score was calculated for each miRNA in the discovery set. This score was based on leave-one-out cross-validation by counting up the results (hit or not) of each leave-out sample and dividing by the number of repeats (the same as total sample number). MiRNAs with cross-validation scores >0.6 were selected.

In the training cohort, the levels of the selected miRNAs were subjected to cluster analysis. Thus, clusters in which the component miRNAs were upregulated or downregulated in a step-by-step manner among healthy, benign, and malignant clusters were identified. Candidate miRNAs were further narrowed down by qRT-PCR analysis of ten malignant and ten benign samples (training set 2). Serum miRNAs that were differentially expressed between two groups (P < 0.05 by Student's t test) were included in the subsequent analysis.

Based on the combinatorial optimization for multicandidate miRNAs, diagnostic indexes were created using Fisher's linear discriminant analysis on the

training set. An index score ≥ 0 was considered to indicate sarcoma, and an index score < 0 to indicate the absence of sarcoma, including the presence of benign bone and soft tissue tumors. Diagnostic sensitivity, specificity, accuracy, PPV, NPV, and AUC were calculated for each selected miRNA marker or combinations of miRNA markers in the training and validation cohorts. Using leave-one-out cross-validation in the training cohort, the best index was finally selected, and several sub-analyses were conducted to verify its clinical utility.

Univariate or age- and sex-adjusted odds ratios were calculated by logistic regression analysis. Discrimination analyses were performed using R version 3.1.2 (R Foundation for Statistical Computing, http://www.R-project.org), compute.es package version 0.2-4, glmnet package version 2.0-3, hash package version 2.2.6, MASS package version 7.3-45, mutoss package version 0.1-10, and pROC package version 1.8. Unsupervised average linkage clustering, heatmap generation with Euclidean distances, and PCA were performed using Partek Genomics Suite 6.6 (Partek, St. Louis, MO, USA). Other statistical analyses, including logistic regression analysis, were performed using IBM SPSS Statistics version 22 (IBM Japan, Tokyo, Japan). A $P$ value < 0.01 was considered statistically significant.

Additional details are included in Supplementary Information.

**Reporting summary**. Further information on experimental design is available in the Nature Research Reporting Summary linked to this article.

## Data availability

All miRNA microarray data and clinical information on the patients who provided the serum samples have been deposited in the Gene Expression Omnibus (GEO) (https://www.ncbi.nlm.nih.gov/geo/) database (accession number: GSE124158). All other relevant data are available within the article file or Supplementary Information, or available from the authors on reasonable request.

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

## Acknowledgements

The study was supported by grants from the Japan Agency for Medical Research and Development, and the Development and New Energy and Industrial Technology Development Organization, Japan. The National Cancer Center Biobank was supported by the National Cancer Center Research and Development Fund, Japan. We thank Tomomi Fukuda, Takumi Sonoda, Hiroko Tadokoro, and Kamakura Techno-Science for performing the microarray assays; Noriko Abe, Michiko Ohori, Kousuke Hirota, Cuneyd Parlayan, Yuuki Tani, and Takumi Sonoda for picking up samples from the freezing room; Kazuki Sudo for independent confirmation of participant eligibility; and Yusuke Yamamoto for critically reading the manuscript.

## Author contributions

Designed the study: N.A., N.T., A.K. and T.O. Performed the experiments and analyzed the data: N.A., J.M., H.S., M.I., J.K., S.T. and Y.A. Contributed materials or clinical data: N.A., A.Y., E.K., Y.T. and A.K. Wrote the manuscript: N.A., J.M., N.T. and T.O. Supervised the study: R.N., H.M., M.M., M.N., T.K., K.K., A.K. and T.O.

## Additional information

**Competing interests:** M.I., J.K., and S.T. are employees of Toray Industries, Inc., the provider of the 3D-Gene® system. Y.A. is an employee of Dynacom Co., Ltd., the developer of the statistical script used to select the best miRNA combination. The remaining authors declare no competing interests.

