## [Peer Review File · Nature Communications]

Reviewers' Comments:

Reviewer #1:

Remarks to the Author:

This is a very interesting and original report addressing the value of miRNA in the peripheral blood to distinguish sarcoma and intermediate risk tumors from benign tumors or other malignancies. This observation and the index described provides an interesting new tool for diagnostic purposes in these rare tumors. Methodology is careful and precise. At this stage, several questions more for the discussion section:

- 1) Do the authors envisage a validation set from a different serum collection?
- 2) Could they propose an algorithm for treatment decision given the PPV, NPV presented for their tool?
- 3) Could they further comment on the value of their index to predict histological subtype since neoadjuvant treatment is the standard for some histotypes.

Reviewer #2:

Remarks to the Author:

Asano, Matsuzaki, Ochiya and colleagues describe development of a microRNA classifier for detection of bone and soft tissue sarcomas using serum samples from ~1000 patients and healthy controls. Using separate discovery, training and validation cohorts, they identify and validate a 3-microRNA index with high diagnostic performance for distinguishing malignant sarcomas from benign disease.

While these findings are exciting and well supported by various analyses of the serum data, my main concern is that the three microRNAs are not found to be cancer-specific in the tissue itself. There are also very weak or no correlation between tissue expression and serum levels of these microRNAs. In addition, there are not differences in serum levels between tumor stage (although not all stages are equally represented in the cohort and this part of the analysis is likely underpowered). These observations raise the question of the true cancer-specificity of these microRNAs and their source in serum?

Nevertheless, lack of an obvious mechanism for release does not detract from discriminatory performance of the serum assays. It does increase the burden of analytical validation though. Could these observed differences have resulted from differences in serum processing or timing of blood collection? How were serum samples processed? What tubes were they collected in? Were they spun down? How long after venipuncture was serum isolated? When were blood samples collected relative to on-going treatments or surgery? It seems healthy volunteers were collected at a different site than cancer patients - were there any differences in protocol between these sites?

Other minor concerns:

- 1) Were the targeted microRNA measurements validated using an orthogonal method?
- 2) What criteria were used to drop samples for "low-quality results"?
- 3) Why did you choose serum instead of plasma for this analysis? Are the targeted microRNA levels concordant in plasma samples (suggesting they are truly extracellular and not just released during cell lysis while clotting and awaiting for serum to separate).

NCOMMS-17-17268

Reply to the Reviewer 1

We would like to thank the reviewer for the constructive comments and suggestions on our manuscript. We made modifications in the Discussion of the manuscript according to the issues raised by the reviewer. Our point-by-point responses are listed below. We hope that the revised version of our manuscript is now suitable for publication in *Nature Communications*.

1) *Do the authors envisage a validation set form a different serum collection?*

Reply: Thank you very much for this important comment. As the reviewer mentioned, this is a crucial point for the design of our study. Although we understand the importance of cohort design, bone and soft tissue tumors are very rare and thus it would be difficult to verify them by adding another retrospective cohort. Furthermore, a considerable proportion of sarcoma patients visited the Rare Cancer Center in the National Cancer Center Hospital from all over the country. This is one of the reasons why it is difficult to create additional sarcoma cohorts from other institutions and hospitals. Therefore, an ideal approach would be to perform a prospective study to verify the usefulness of the index II as a sarcoma biomarker. We have just started this study. We have explained the situation in the Discussion section (page 14, line 306 - 308).

2) *Could they propose an algorithm for treatment decision given the PPV, NPV presented for their tool?*

Reply: In actual clinical practice, Index II would be useful for the differential diagnosis of benign and malignant bone and soft tissue tumor, or differential diagnosis of sarcoma and other cancers, especially in cases where differential diagnosis is difficult. Furthermore, we consider that the index II could also be applied as a screening tool for sarcomas during medical examinations. We have added a possible clinical application of Index II to the Discussion (page 14, line 302 - 306).

3) *Could they further comment on the value of their index to predict histological subtype since neoadjuvant treatment is the standard for some histotypes?*

Reply: As the reviewer mentioned, neoadjuvant chemotherapy is a standard treatment for some

histological subtypes, such as osteosarcoma, Ewing sarcoma, and rhabdomyosarcoma. We also think that the identification of diagnostic markers that can distinguish between these tissue types is a very attractive but challenging prospect. In this study, Diagnostic Index II (and serum miRNA profiles) could not distinguish tumors by histological subtype. This is ascribed to the fact that sarcomas share a common mesenchymal origin. Therefore, although serum miRNAs have potentially the ability to classify specific histological subtypes of sarcomas, it would be very difficult to design a study using large numbers of serum samples from individual sarcomas because of the rarity of this disease. We have added a discussion about to the Discussion (page 13, line 293 – page 14, line 301).

NCOMMS-17-17268

Reply to the Reviewer 2

We would like to thank the reviewer for the constructive comments and suggestions on our manuscript. We modified Methods and Discussion according to the comments and suggestions raised by the Reviewer. We hope that the Reviewer will find the new version satisfactorily revised and acceptable for publication in *Nature Communications*. Our point-by-point responses are listed below.

1) While these findings are exciting and well supported by various analyses of the serum data, my main concern is that the three microRNAs are not found to be cancer-specific in the tissue itself. There are also very weak or no correlation between tissue expression and serum levels of these microRNAs. In addition, there are not differences in serum levels between tumor stage (although not all stages are equally represented in the cohort and this part of the analysis is likely underpowered). These observations raise the question of the true cancer-specificity of these microRNAs and their source in serum?

Reply: We totally agree with your opinion. We also do not believe that all of the identified serum miRNAs were secreted directly from tumor cells into the circulation. Conventional tumor markers, such as PSA and CEA, are specifically released from tumor cells. By contrast, since circulating miRNAs can be released not only from tumor cells but also from the other various kind of cells, the specific profiles of serum miRNAs in sarcoma patients will not always reflect the profiles of miRNAs released from sarcoma cells. In fact, as shown in supplemental Table 3, the three miRNAs identified were found in both malignant and benign groups, although their levels were significantly different. This indicates that these miRNAs were probably not specifically released from sarcoma cells but released from other cell types. We speculate that the presence of a sarcoma might affect the mesenchymal cells in its vicinity, such as fibroblasts, macrophages, lymphocytes, and endothelial cells. In this study, we were able to detect the tumor-bearing condition, including tumor-associated changes in various cells, using serum miRNA levels. Serum miRNA profiles are considered to mirror the specific changes that occur in cancer patients, such as inflammation and altered immune responses. Currently, the origins and functions of these three serum miRNAs are unknown. It will be of considerable interest in the future to determine the functions of these sarcoma-specific circulating miRNAs and how they are related to the development and progression of sarcomas. We have added a discussion about this point to the revised version of the manuscript (page 13, line 283 - 292).

2) Nevertheless, lack of an obvious mechanism for release does not detract from discriminatory performance of the serum assays. It does increase the burden of analytical validation though. Could these observed differences have resulted from differences in serum processing or timing of blood collection? How were serum samples processed? What tubes were they collected in? Were they spun down? how long after venipuncture was serum isolated? When were blood samples collected relative to on-going treatments or surgery? It seems healthy volunteers were collected at a different site than cancer patients - were there any differences in protocol between these sites?

Reply: Serum samples were collected for clinical purposes, and surplus sera after laboratory testing were stored at -20°C at the National Cancer Center Biobank. According to usual clinical procedures, whole serum samples are centrifuged and stored within 12 hours after collection using serum collection tubes (VP-AS109K50 or VP-AR109K63, TERUMO, Tokyo, Japan). In our study, pre-treatment serum samples were obtained before surgery or chemotherapy, and samples from patients who had undergone treatments such as chemotherapy or radiotherapy were excluded. Although the sampling process, such as the time interval between centrifugation and storage, was not strictly regulated and differed between samples, this would not have been expected to create bias when comparing malignant and benign samples because both types of samples were stocked at the same biobank.

MiRNAs are reported to more stable in collected sera than mRNAs or proteins (Mitchell, et al. PNAS. 2008.). In fact, we could still detect miRNAs even after long-term frozen. In addition, we validated our results using an independent sample set, although the handling processes were not matched. This suggests that our results were robust irrespective of the sampling procedures, although further prospective validations are needed.

In contrast to samples from malignant patients, samples from healthy controls were collected at a different site and stocked in -80°C. We agree that direct comparison between our malignant samples and the healthy control samples could lead to some bias in the results. Therefore, we did not select biomarker miRNA candidates by comparing miRNAs between malignant and healthy samples. Rather, we used healthy samples to further select the miRNA candidates that were upregulated or downregulated both in benign and healthy samples as compared with malignant samples. This process allowed us to exclude some miRNAs whose serum levels were altered only in patients with benign tumors but not in patients with malignant tumors.

The following information has been added to Methods: “All serum samples were frozen at -20°C

within 12 hours of collection using serum collection tubes (VP-AS109K50 or VP-AR109K63, TERUMO, Tokyo, Japan) (page 15, line 324 - 326)

Minor concerns:

1) *Were the targeted microRNA measurements validated using an orthogonal method?*

Reply: Thank you very much for this important comment. We attempted to evaluate the levels of the three miRNAs (miR-1292-3p, miR-8071, miR-3195) for the diagnosis of sarcoma, as well as those of internal-control miRNAs (miR-149-3p, miR-2861 and miR-4463), by quantitative RT-PCR (qRT-PCR) using the TaqMan® microRNA assay (Thermo Fisher Scientific, San Jose, CA, USA). However, specific primer sets for miR-3195, miR-2861 and miR-4463 could not be obtained commercially available because of problems encountered in primer design.

We randomly selected samples from 10 patients with malignant tumors and 10 patients with benign tumors. Then, the levels of miR-1292, miR-8071, and miR-149-3p were evaluated in the 20 samples using qRT-PCR. However, Ct values for the three miRNAs were ~35 (mean Ct value [range]; miR-1292, 33.7 [31.3-37.0]; miR-8071, 34.4 [32.2-37.8]; miR-149-3p, 34.6 [31.9-38.4]), showing that the serum levels of these miRNAs could not be evaluated using conventional qRT-PCR. In fact, no differences in the levels of miR-1292 and miR-8071 (normalized by miR-149-3p) were found between malignant and benign samples as shown in the following figure.

We have experienced this situation quite often in qRT-PCR analysis of serum miRNAs. This is why we used 3D-Gene® microarray analysis, which is known to be highly sensitive. Based on the present results, we are now considering the development of clinical applications for these miRNA markers based on using the same microarray platform. We have started a clinical prospective validation study, but obtaining results may take at least two years since sarcomas are rare.

2) *What criteria were used to drop samples for "low-quality results"?*

Reply: We apologize for the confusing description. The quality control process for microarray analysis is described in “MiRNA expression array” in Methods as follows: “The microarray data was subjected to quality control using a coefficient of variation for the negative control probes of <0.15 and the number of flagged probes of <10. Flagged probes mean uneven hybridization, which was automatically identified by the 3D-Gene® Scanner” We further added an explanation for “flagged probes” as follows: “Flagged probes mean uneven hybridization, which were automatically identified by 3D-Gene® Scanner.” (page 16, line 354 - 356)

3) *Why did you choose serum instead of plasma for this analysis? Are the targeted microRNA levels concordant in plasma samples (suggesting they are truly extracellular and not just released during cell lysis while clotting and awaiting for serum to separate).*

Reply: We used serum samples because all the blood samples in the NCC biobank are serum samples. As you mentioned, plasma miRNA levels should reflect the actual amount of circulation miRNAs better than serum levels. Some reports have shown that miRNA levels correlate well between plasma and serum (Wang, et al. PLoS ONE 2012, Mitchell, et al. PNAS. 2008.). Serum collection is more generally performed in clinical practice than plasma collection. Therefore, serum biomarkers in serum would be beneficial since we can conduct both laboratory tests and miRNA evaluation all at once.

Reviewers' Comments:

Reviewer #1:

Remarks to the Author:

No further comments

Reviewer #2:

Remarks to the Author:

Asano, Matsuzaki, Ochiya and colleagues describe development of a microRNA classifier for detection of bone and soft tissue sarcomas using serum samples from ~1000 patients and healthy controls. Using separate discovery, training and validation cohorts, they identify and validate a 3-microRNA index with high diagnostic performance for distinguishing malignant sarcomas from benign disease.

Unfortunately, the revised version has not completely addressed concerns raised previously:

1) Lack of orthogonal validation is significantly concerning. The authors were unable to validate even one of the 3 candidate microRNAs presented using qRT-PCR. It is unclear why primer design failed in some cases.

2) Further, it is unclear how much extracted RNA / equivalent serum volume was used for the microarray analysis and the validation qRT-PCR. Without orthogonal validation, the results of this study are less convincing.

3) What is the distribution of potential confounding variables between patients with malignant and benign disease and healthy volunteers? It does not seem the analysis was adjusted for these confounders? The p-values presented in Table 1 compare confounders between discovery and validation cohorts but not between test groups.

4) The abstract states "Circulating serum miRNA profiles in sarcoma patients were clearly distinct from those in patients with other types of tumors", although not enough detail is provided about this comparison with other tumor types in the manuscript. How were these tumors selected, since it seems 30 cases of each tumor type were used.

5) The authors have not commented on differences between the diagnostic signature between tumor stage, size etc. It seems the actual size and invasiveness of the tumor (T-stage) makes no difference to diagnostic index II levels – disagreeing with the authors' hypothesis that tumor causes disruption of the surroundings and leads to release of these microRNAs

6) Superlatives like 'revolutionary new strategy' are not helpful. The use of microRNAs for early detection of tumors is not a novel strategy in itself and the manuscript needs to be revised accordingly.

7) Conclusion seems to have a typo, suggesting only 2 microRNAs are part of the final diagnostic signature instead of 3.

Response to Reviewers' comments:

In the re-revised version of the manuscript, we carefully addressed all of the comments raised by the reviewers, and revised or added sentences to the different sections according to the reviewers' comments. The orthogonal validation using miRNA sequencing was performed, and the results were added to the Results and Methods sections and Supplemental Table 6 and Figure 6. The serum volumes used for microarray analysis and miRNA sequencing were included in the Methods section. The distributions of age and sex between test groups were described in the Results section and Supplemental Table 3. The selection of other cancer samples was described in the Methods section. We also added a discussion describing the fact that Index II did not increase according to TNM stage. Other minor changes were performed according to the reviewer's comments. A list of our point-by-point responses is also enclosed. We hope that the re-revised version of the manuscript is now acceptable for publication in *Nature Communications*.

Reply to Reviewer 2

We would like to thank the reviewer for the constructive comments and suggestions regarding our manuscript. We modified the manuscript according to the comments and suggestions of the reviewer. We hope that the reviewer will find the new version satisfactorily re-revised and acceptable for publication in *Nature Communications*. Our point-by-point responses are listed below.

Asano, Matsuzaki, Ochiya and colleagues describe development of a microRNA classifier for detection of bone and soft tissue sarcomas using serum samples from ~1000 patients and healthy controls. Using separate discovery, training and validation cohorts, they identify and validate a 3-microRNA index with high diagnostic performance for distinguishing malignant sarcomas from benign disease.

Unfortunately, the revised version has not completely addressed concerns raised previously:

- 1) Lack of orthogonal validation is significantly concerning. The authors were unable to validate even one of the 3 candidate microRNAs presented using qRT-PCR.*

Reply:

For the orthogonal validation, we performed miRNA sequencing (miRNA-Seq) using 20 serum samples (10 sarcoma and 10 benign tumors) that were randomly selected from the validation set (Supplemental Table 6). The amount of total RNA used for construction of the sequencing library was twice that used for microarray analysis (miRNA-Seq, 300 μ L of serum; microarray, 150 μ L of serum). Sequence reads consistent with mature miRNA sequences according to miRBase release 21 were counted. When we defined the full length of a mature miRNA sequence as 'L', the perfect match in length of 'L-2' was counted as the presence of this miRNA. The sequence variations of miRNAs, such as 5' trimming, 3' trimming, substitutions, and 3' additions, were reported previously (Front Genet 6:186, 2015), and the signal levels of microarray reflect the sum amount of the mature sequence and the variants of each miRNA.

Unsupervised clustering analysis was used to validate differences in circulating miRNA profiles between primary sarcomas and benign tumors (Supplemental Fig. 6A). A significant correlation between the microarray and miRNA-Seq results was observed for miRNAs with read counts greater than 100 (Supplemental Fig. 6B). However, for miRNAs with limited read counts (≤ 100), no correlation was observed between the microarray and miRNA-Seq results despite signal levels of $>2^6$ for some miRNAs based on microarray results. This suggests that microarray is more sensitive for the detection of serum miRNAs than miRNA-Seq.

Subsequently, we focused on the read counts of 11 miRNAs showing significant differences between sarcomas and benign tumors based on the microarray results of the discovery set. Except for three miRNAs (miR-1273g-3p, miR-3195, and miR-4258), sequence reads of eight miRNAs were not detected in any of the 20 samples. Microarray results showed that the signal levels of the three detectable miRNAs were comparably high among the 11 miRNAs (Supplemental Fig. 2). The read counts of the three miRNAs were higher in the serum of patients with sarcoma than in that of patients with benign tumors, which was consistent with the results of microarray (Supplemental Fig. 6C).

We therefore successfully confirmed the accuracy and high sensitivity of our microarray platform. In addition, although the levels of miR-1292-3p and miR-8071 in sera could not be evaluated using miRNA-Seq, the upregulation of miR-3195 in sarcoma was validated. To validate the results of miR-1292-3p and miR-8071, a considerably greater amount of serum needs to be used for miRNA-Seq. However, we do not have enough residual serum samples for further validation.

These results are described in the Results section (pages 11–12, lines 231–255), Methods section (page 18-19, lines 415-427), Supplemental Table 6, and Figure 6.

2) *It is unclear why primer design failed in some cases.*

Reply:

Among 2565 mature miRNA sequences, the detection probes for 211 miRNAs are not listed in the catalog of the TaqMan MicroRNA Assay platform (Thermo Fisher). The detection probes for 94 miRNAs are not listed in the catalog of the TaqMan Advanced miRNA Assay platform (Thermo Fisher). The detectability of miRNAs using qRT-PCR is dependent on the sequences, such as high GC contents or the potential for self-dimerization. In the case of miR-3195, neither qPCR platforms were available because of the short length (only 17 nt) and extremely GC rich sequence (all GCs except the last 2 nt), according to the manufacturers' answer to our question.

In addition, according to our miRNA-Seq results, the number of sequence reads that were perfectly matched with mature miRNA sequences based on miRBase was limited, as indicated by the yellow boxes in Supplemental Figure 6C. For PCR quantification, the sequences need to be perfectly matched to the primer design. Therefore, we often find it difficult to validate the results of microarray using qRT-PCR platforms.

3) *Further, it is unclear how much extracted RNA / equivalent serum volume was used for the microarray analysis and the validation qRT-PCR. Without orthogonal validation, the results of this study are less convincing.*

Reply:

One half of the amount of total RNAs extracted from 300 μ L of serum was subjected to microarray analysis based on the standard operating procedure in our project. On the other hand, the whole amount of total RNAs extracted from 300 μ L of serum was subjected to miRNA-Seq. This point was described in the Methods section of the revised manuscript (page 17, lines 393–394, and page 19, lines 416–419).

The orthogonal validation was performed using miRNA-Seq and has been described above.

4) *What is the distribution of potential confounding variables between patients with malignant and benign disease and healthy volunteers? It does not seem the analysis was adjusted for these confounders? The p-values presented in Table 1 compare confounders between discovery and validation cohorts but not between test groups.*

Reply:

We described the differences in age and sex between participants with malignant/benign/intermediate tumors and healthy volunteers in the Results section (pages 6–7, lines 127–130), and Supplemental Table 3. In addition, significant detectability of sarcoma by Index II was conserved after adjustment for age and sex, as shown in the Results section (page 9, lines 180–181) and Supplemental Table 5.

- 5) *The abstract states “Circulating serum miRNA profiles in sarcoma patients were clearly distinct from those in patients with other types of tumors”, although not enough detail is provided about this comparison with other tumor types in the manuscript. How were these tumors selected, since it seems 30 cases of each tumor type were used.*

Reply:

We apologize for the insufficient description. We described this in the Methods section (page 17, lines 377–381) as follows:

“Serum samples were obtained from patients with eight types of cancer (breast, brain, pancreatic, colorectal, hepatic, esophageal, gastric, and lung cancers; n=30 for each) who were randomly selected among patients admitted or referred to the NCCCH between 2008 and 2013, and registered in the National Cancer Center Biobank. Serum samples were stored at –20°C until use.”

- 6) *The authors have not commented on differences between the diagnostic signature between tumor stage, size etc. It seems the actual size and invasiveness of the tumor (T-stage) makes no difference to diagnostic index II levels – disagreeing with the authors’ hypothesis that tumor causes disruption of the surroundings and leads to release of these microRNAs.*

Reply:

Thank you very much for this important comment. As the reviewer mentioned, the Index II value did not increase according to the tumor burden unlike conventional tumor markers, such as PSA and CEA, which are specifically released from tumor cells. This feature can be explained as follows.

First, the three miRNAs constituting Index II are not all tumor-secreted miRNAs. In fact, miR-8071 was the only miRNA showing a correlation between expression in tumor tissue and serum (Supplementary Fig. 5). In addition, the Index II value is derived from a formula designed for

discriminating between benign and malignant tumors using a combination of three miRNAs. This is a distinctive feature that differentiates it from tumor-secreted conventional biomarkers. These features do not necessarily decrease the utility of Index II because a good performance of the index for discriminating between benign and malignant tissues does not imply that the value increases with tumor burden. We added a discussion about this point to the revised version of the manuscript (pages 13–14, lines 296–305).

7) *Superlatives like ‘revolutionary new strategy’ are not helpful. The use of microRNAs for early detection of tumors is not a novel strategy in itself and the manuscript needs to be revised accordingly.*

Reply:

Thank you for your suggestion. As pointed out, ‘revolutionary new strategy’ might be an overstatement. We changed this to a simple sentence in the abstract and introduction section (page 3, line 52-53, page 6, lines 110–111) as follows:

“Our findings provide a new solution to the important problems associated with the diagnosis of sarcoma.”

7) *Conclusion seems to have a typo, suggesting only 2 microRNAs are part of the final diagnostic signature instead of 3.*

Reply:

We apologize for this error. As pointed out, the sentence should read three miRNAs and not two miRNAs. We corrected this in the Discussion section (page 16, line 352).

Reviewers' Comments:

Reviewer #2:

Remarks to the Author:

Asano, Matsuzaki, Ochiya and colleagues describe development of a microRNA classifier for detection of bone and soft tissue sarcomas using serum samples from ~1000 patients and healthy controls. Using separate discovery, training and validation cohorts, they identify and validate a 3-microRNA index with high diagnostic performance for distinguishing malignant sarcomas from benign disease.

1. In this revised version, the authors presented validation data using miRNA sequencing of 20 cases (10 malignant and 10 benign). However, not enough detail of miRNA sequencing was provided. There was no indication of how much sequencing data was generated per sample, how much of that was on-target to provide miRNA coverage. I am also intrigued why they pursued 100-bp reads for a miRNA sequencing run?

While overall read count is not reported, only 3 of 11 miRNA of interest are investigated directly in the miRNA sequencing validation run with only 1 of the 3 miRNA that are part of the final signature. The other 8 were not detected even once in any of the 20 serum samples, suggesting there is much more room for deeper sequencing analysis to pursue this orthogonal validation set.

2. In the revised version, the authors added age and gender-adjusted odds ratios for the signature in Supplementary Table 5. However, there are no details on what kind of statistical analysis was used for this adjustment. They don't provide any details on whether this is true for comparison between malignant and benign tumors and only focus on the validation set (no information provided for the training set). If indeed the signature and its components retain diagnostic power after age and gender-adjustment, this aspect of the finding would merit reporting clearly and upfront – it will make this paper stronger.

Response to Reviewers' comments:

We have carefully addressed all of the comments raised by the reviewers, and have either revised the text or added more information as requested. Our point-by-point responses to each of the comments are attached below. All changes to the manuscript are highlighted in yellow.

NCOMMS-17-17268C

Reply to Reviewer 2

We thank the reviewer for the constructive comments and suggestions regarding the manuscript. We have revised the manuscript accordingly.

***Reviewers' comments:** Reviewer #2 (Remarks to the Author): Asano, Matsuzaki, Ochiya and colleagues describe development of a microRNA classifier for detection of bone and soft tissue sarcomas using serum samples from ~1000 patients and healthy controls. Using separate discovery, training and validation cohorts, they identify and validate a 3-microRNA index with high diagnostic performance for distinguishing malignant sarcomas from benign disease.*

1-1)

In this revised version, the authors presented validation data using miRNA sequencing of 20 cases (10 malignant and 10 benign). However, not enough detail of miRNA sequencing was provided. There was no indication of how much sequencing data was generated per sample, how much of that was on-target to provide miRNA coverage.

Reply:

We apologize for the lack of detailed description. We have now described miRNA sequencing in the Results section (page 12, line 238-240) as follows:

“On average, 6.87 million (range: 4.96–8.78 million) reads were generated per sample; the average number of miRNA reads was 818 (range: 97–3730) per sample.”

In addition, we apologize for analyzing raw miRNA-Seq data. We have repeated the analysis using

normalized data (read counts per 10^6 total reads). The results were no different from those included in the previous version of revised manuscript.

I-2)

I am also intrigued why they pursued 100-bp reads for a miRNA sequencing run?

Reply:

We did this because we followed the specifications of the sequencing service provider (Takara Bio Inc.). As you suggest, 50 bp sequencing would be sufficient to identify adapter sequences with a view to distinguishing sample sources and trimming the reads. However, we note that some previous studies also performed 100 bp sequencing for small RNA profiling (BMC Med Genomics. 2018, 11:48; FEBS Lett. 2016, 590:3574). We cannot specify the lowest number of reads needed for appropriate analysis as the adapter sequences are not available publically.

I-3)

While overall read count is not reported, only 3 of 11 miRNA of interest are investigated directly in the miRNA sequencing validation run with only 1 of the 3 miRNA that are part of the final signature. The other 8 were not detected even once in any of the 20 serum samples, suggesting there is much more room for deeper sequencing analysis to pursue this orthogonal validation set.

Reply:

As the reviewer points out, we did not validate 8 of the 11 miRNAs of interest. In the revised version, we have provided the overall read count per sample (as mentioned above). The total read counts were around 7 million, which means that we succeeded in obtaining sufficient sequencing data. The main reason for the limited number of miRNA read counts was the low amount of RNA (1.2–2.2 ng per sample). Unfortunately, we did not have enough residual serum samples for sequencing. To resolve this, we amplified small RNAs using the SMARTer system (Takara) (which is PCR-based) for library preparation. PCR amplification enabled us to sequence small amounts of small RNAs; however, the amplification efficiency was not consistent among miRNA sequences. Therefore, some miRNA sequences can be difficult to read using a limited amount of RNA. Notably, the microarray signal levels of the three detected miRNAs were comparatively high among the 11 miRNAs. Thus, it appears that this problem cannot be resolved simply by undertaking deeper sequencing analysis. We understand the importance of orthogonal validation for this type of study; however, we think that it

is a general problem in the field of miRNA biomarker research. As supported by the miRNA sequencing data, variations at both ends of the miRNA are extremely important if we are to correctly evaluate the amount of circulating miRNAs. We have begun to create a new strategy for evaluating variations in each extracellular miRNA using a limited amount of RNA from a sample. We also believe that these results indicate that our microarray analysis is a more sensitive tool than miRNA sequencing when using small amounts of serum.

2-1)

In the revised version, the authors added age and gender-adjusted odds ratios for the signature in Supplementary Table 5. However, there are no details on what kind of statistical analysis was used for this adjustment.

Reply:

We apologize for the lack of detail. We have revised the Methods section (page 23, line 465-466) as follows:

“Univariate or age- and sex-adjusted odds ratios were calculated by logistic regression analysis.”

2-2)

They don't provide any details on whether this is true for comparison between malignant and benign tumors and only focus on the validation set (no information provided for the training set). If indeed the signature and its components retain diagnostic power after age and gender-adjustment, this aspect of the finding would merit reporting clearly and upfront – it will make this paper stronger.

Reply:

Thank you for the constructive suggestions. Accordingly, we have revised Supplementary Table 5. Age- and sex-adjusted odds ratios (Index II), as well as their components in both the training and validation sets, are described.

Reviewers' Comments:

Reviewer #2:

Remarks to the Author:

Unfortunately the miRNA sequencing details show on average, 0.01% of the sequencing reads mapped to miRNAs. This suggests miRNA sequencing may have failed, considering published literature would suggest ~20% mapping rate for plasma-derived miRNAs.

(<https://www.frontiersin.org/articles/10.3389/fgene.2013.00020/full>)

Even if the on-target rate is expected for the specific prep the authors used for miRNA prep, the number of reads per miRNA species evaluated is low (total 818 miRNA reads on average, per sample). This suggests imprecise quantification and it is unlikely to be meaningful as quantitative orthogonal validation.

Re: NCOMMS-17-17268C

Reply to Reviewer #2

We thank the reviewer for their constructive comments and suggestions regarding orthogonal validation. We have revised the manuscript accordingly.

Reviewers' comments:

Unfortunately the miRNA sequencing details show on average, 0.01% of the sequencing reads mapped to miRNAs. This suggests miRNA sequencing may have failed, considering published literature would suggest ~20% mapping rate for plasma-derived miRNAs. (<https://www.frontiersin.org/articles/10.3389/fgene.2013.00020/full>)

Even if the on-target rate is expected for the specific prep the authors used for miRNA prep, the number of reads per miRNA species evaluated is low (total 818 miRNA reads on average, per sample). This suggests imprecise quantification and it is unlikely to be meaningful as quantitative orthogonal validation.

Reply:

We appreciate this thoughtful comment. We completely agree that orthogonal validation is important. Unfortunately, in our original manuscript, we did not clearly validate the quantity of the three microRNAs of Index II using qRT-PCR as well as miRNA-seq. Because the serum miRNA profiles themselves were clearly distinct between sarcomas and benign tumors, we changed our strategy of miRNA selection (Fig. 1) and reconstructed the diagnostic biomarker, which could then be validated by qRT-PCR. We believe that our manuscript has been greatly improved due to your comments.

Specifically, we changed the following four methods and reanalyzed accordingly:

1) First, to select more robust biomarkers, we changed the method for the selection of miRNAs from fold change and *P* value to the leave-one-out cross-validation method in the discovery group. Based on the new method, 83 miRNAs exhibited high discrimination performance between sarcomas and benign tumors in the discovery set (Supplemental Table 4).

2) Second, suitable miRNAs were further selected in the training cohort. According to cluster analysis, 33 of these 83 miRNAs were included in a cluster in which expression increased through healthy, benign, and malignant (sarcoma) groups in a step-by-step manner (Supplemental Fig. 2).

3) Third, we performed qRT-PCR analysis using 10 malignant and 10 benign samples (training set 2, shown in Table 1, Fig. 1, and Supplemental Table 3) using the miScript PCR System (QIAGEN, Hilden, Germany). Among the 33 miRNAs referred to above, miR-4488, miR-1908-3p, miR-1292-3p, miR-6088, and miR-4492 were excluded because specific primers for use in the miScript PCR System were not commercially available. According to the results of qRT-PCR for 28 miRNAs, we further selected 12 miRNAs whose expression levels were higher in malignant samples than in benign samples ($P < 0.05$) (Supplemental Fig. 3).

4) As in the previous version of the manuscript, we calculated diagnostic indices using combinations of serum miRNAs in the training cohort. The best indices for combinations of two to nine miRNAs, obtained using cross-validation in the training cohort, are shown in the lower row of Table 2. In this revision, a combination of seven miRNAs (Index VI) exhibited the highest accuracy with the lowest number of miRNAs (Table 2). The ability of Index VI to detect sarcoma was preserved after adjustment for age and sex (Supplemental Table 5). Therefore, we chose Index VI for further validation as a biomarker for the detection of malignant sarcomas. The ROC curves of each of the seven miRNAs, as well as Index VI itself, are shown in Fig. 2E and F.

Finally, we confirmed that Index VI was not inferior to Index II from the previous manuscript as a biomarker for the detection of malignant sarcomas (Fig. 3, Supplemental Fig. 4–6).

Based on these considerations, we made major changes to the Results, Discussion, and Methods sections, Table 1, Figs. 1, 2C–F, and 3, Supplemental Tables 1 and 3–5, and Supplementary Figs. 2–6. Other minor changes were also made. All changes to the manuscript are highlighted in yellow.

Using qRT-PCR, we successfully orthogonally validated Index VI obtained in this revised version. Therefore, we are convinced that it is a more robust and reliable biomarker than the previous Index II.

Reviewers' Comments:

Reviewer #2:

Remarks to the Author:

Asano, Matsuzaki, Ochiya and colleagues describe development of a microRNA classifier for detection of bone and soft tissue sarcomas using serum samples from ~1000 patients and healthy controls. Using separate discovery, training and validation cohorts, they identify and validate a 7-microRNA index with high diagnostic performance for distinguishing malignant sarcomas from benign disease.

This version has been substantially revised and addresses prior concerns, particularly that of orthogonal validation. I am surprised that the diagnostic signature has changed so dramatically but as it stands now, the authors have done a great job validating their findings as much as possible within a single study. The conclusions are reasonably presented and supported by their data.

Re: NCOMMS-17-17268D

Reply to Reviewer #2

Comments:

Asano, Matsuzaki, Ochiya and colleagues describe development of a microRNA classifier for detection of bone and soft tissue sarcomas using serum samples from ~1000 patients and healthy controls. Using separate discovery, training and validation cohorts, they identify and validate a 7-microRNA index with high diagnostic performance for distinguishing malignant sarcomas from benign disease.

This version has been substantially revised and addresses prior concerns, particularly that of orthogonal validation. I am surprised that the diagnostic signature has changed so dramatically but as it stands now, the authors have done a great job validating their findings as much as possible within a single study. The conclusions are reasonably presented and supported by their data.

Reply: We would like to thank the reviewer for the favorable comments regarding our revised manuscript. According to your constructive opinions, we could develop more useful and robust serum biomarkers of sarcoma.